# Unveiling the Drivers of Chinese Tourists' Visit Intentions Regarding Malaysia

**Xiaocong Jiang [1,2,\*]** , **Ahmad Edwin bin Mohamed [2]** and **Amirul Husni bin Affifudin [2]**

[1]  School of Business, Institute of Vocational Technology, SIP—Suzhou Industrial Park, Suzhou 215123, China
[2]  Ghazali Shafie Graduate School of Government, Northern University of Malaysia, Sintok 06010, Malaysia; edwin@uum.edu.my (A.E.b.M.); amirulhusni@uum.edu.my (A.H.b.A.)
\*   Correspondence: roy@ivt.edu.cn

**Abstract:** In 2023, the number of Chinese tourists visiting Malaysia had not yet returned to pre-pandemic levels, unlike those from some regions where tourism numbers have normalized. The lack of established research methodologies complicates the determination of whether negative news reports contribute to reduced visit intentions among these tourists. Through semi-structured interviews with a total of 69 individuals, including tourists who have visited, those who planned to visit but canceled, and local industry professionals, and using thematic analysis, this study identified ten primary factors diminishing Chinese tourists' visit intentions regarding Malaysia. Notably, the findings suggest that the main reasons are not primarily associated with negative media coverage. The research indicates that improvements in multilingual services, targeted marketing strategies, effective use of Chinese social media platforms, promotion of local culture, addressing inaccuracies in religious and cultural guidance, and reducing regional disparities in infrastructure could enhance the visit intentions of Chinese tourists regarding Malaysia. This study not only offers a comprehensive framework for understanding the factors influencing visitation intentions but also provides an effective methodology for assessing the impact of unforeseen events on tourist behavior. It further proposes practical strategies to enhance the recovery of tourist arrivals.

**Keywords:** Chinese tourists; Malaysia; visit intention; semi-structured interviews; thematic analysis

## 1. Introduction

In 2023, the global tourism sector began recovering from the disruptions caused by the COVID-19 pandemic. A key area of focus for international tourism destinations has been understanding 'visit intention'—the desire and planning of individuals to travel to specific destinations. Notably, China has emerged as a global leader in the outbound tourism market, leading in both travel frequency and spending [1]. According to ref. [2], Chinese outbound tourism numbers reached 87 million in 2023, achieving 60% of the pre-pandemic levels recorded in 2019. An overwhelming 93.95% of these tourists expressed a preference for visiting neighboring countries and regions, with Malaysia being a prominent preferred destination [3]. Ref. [4] reported that China ranked third in contributing to Malaysia's inbound tourism, accounting for 11.9% of total arrivals and generating 17.8% of Malaysia's tourism-related revenue prior to the COVID-19 pandemic. According to ref. [5], the largest group of international visitors to Malaysia came from Singapore, followed by Indonesia, with China being the third largest source. In the year before the pandemic, 2019, visitors from Singapore spent an average of RM 2021.6 each, reflecting a decrease of 21.3% from 2018. Conversely, Chinese tourists demonstrated a higher spending capacity, with an average expenditure of RM 4921.0 per person, marking an increase of 17.7% from the previous year. This spending pattern was also evident in overall tourism revenue, as Singaporean tourists contributed RM 20,547.3 million to Malaysia's tourism industry in 2019, which was a decrease of 24.6% from the previous year. Although fewer in number

compared to Singaporean visitors, Chinese tourists made a substantial contribution of RM 15,325.3 million, an increase of 24.5% from the previous year. Furthermore, before the pandemic, Chinese tourists were also the largest source of tourism expenditure in Japan and Australia [1]. Studying the visit intentions of Chinese tourists can positively affect destination countries that consider China to be one of their main sources of tourists after the pandemic. But, the number of Chinese tourists traveling to Malaysia in 2023 represented only 30.1% of the figure from the corresponding period in 2019. In contrast, data from ref. [6] indicate that the numbers of tourists from Singapore and Indonesia to Malaysia have rebounded to 75% and 80% of the pre-pandemic levels, respectively.

In 2023, the film "No More Bets," which generated significant discussion within Chinese-language social media circles, spotlighted the experiences of Chinese tourists on Malaysian islands, focusing on controversial issues such as cybercrime, human trafficking, and illegal organ trading [7]. Online debates were also spurred by reports of exploitation involving Malaysian immigration officials and unethical behavior by local tourism workers towards Chinese visitors [8,9]. Ref. [10] expressed concern about the potential link between the failure of Chinese tourist arrivals to return to pre-pandemic levels and the aforementioned sudden negative news events. These concerns, while deeply troubling, leave us in a state of uncertainty regarding their potential impact on the intentions of Chinese tourists to visit. The visit intentions of Chinese tourists have significantly changed since the period prior to the pandemic [1], and such changes may have intensified after the pandemic [11].

However, current research on destination visit intention exhibits several deficiencies. Firstly, the impact of negative news on tourists' visit intentions through induced negative emotions remains a contentious topic within academia. Traditional methodologies such as surveys and User-Generated Content (UGC) analysis face limitations in addressing sudden negative events. Surveys may not effectively uncover tourists' emotional responses [12], and UGC may suffer from data scarcity when dealing with unexpected incidents, limiting its accuracy and validity [13]. Consequently, there is a lack of effective methods to accurately analyze whether sudden negative news has elicited negative emotions sufficient to impact visit intention, making it challenging to identify the primary factors influencing tourists' intentions to visit.

Secondly, although Crouch and Ritchie [14] have highlighted that tourism image, attractiveness, and competitiveness significantly influence visit intention, the existing literature often focuses on analyzing a single dimension (e.g., [15–17]). This narrow focus may not fully capture the multifaceted nature of visit intention, which is influenced by multiple factors, thus preventing a deeper understanding of tourist choice behavior. Furthermore, the majority of studies tend to analyze visit intention towards Malaysia through specific types of tourism (e.g., urban, rural, and coastal tourism) [18–20] or particular tourism projects (e.g., sports and cultural tourism) [21,22], with less of the literature adopting a broader perspective that considers Malaysia as a whole.

Thirdly, to the best of our knowledge, the few studies exploring post-pandemic visit intentions to Malaysia are based on pre-pandemic data or focus solely on Muslim tourists [23,24], thereby preventing a comprehensive understanding of current visit intentions among Chinese tourists.

We conducted semi-structured interviews with 69 Chinese tourists who visited Malaysia in 2023, those who had planned but ultimately canceled their trips, and local tourism service providers and managers. The data collected from these interviews were analyzed using thematic analysis. By integrating the three core dimensions of tourism image, attractiveness, and competitiveness, we constructed a comprehensive framework to analyze the factors influencing destination choice and visit intentions. This framework aims to identify the main factors affecting tourists' intention to visit and their underlying motivations, as well as to determine whether recent sudden negative news on Chinese social media is the main reason for the decline in visit intentions.

This study seeks to answer the following research questions (Table 1):

**Table 1.** Research questions and justification of relevance based on references.

| Research Question | References |
|---|---|
| Q1: In light of recent sudden negative news, what are the key factors influencing the reduction in Chinese tourists choosing Malaysia as a travel destination, and are there more reliable research methods to effectively explore these factors? | [12–14] |
| Q2: Considering multiple dimensions such as tourism image, attractiveness, and competitiveness, and viewing Malaysia as a whole, what factors significantly affect the intention of Chinese tourists to visit to Malaysia after the pandemic? | [15–17,23,24] |

## 2. Literature Review

To comprehensively understand the multidimensional factors influencing Chinese tourists' intentions to visit Malaysia, this study employed a systematic literature review method, strictly adhering to the PRISMA guidelines to ensure high transparency and replicability. This study conducted extensive searches from 2010 to 2023 in major databases like Google Scholar, Web of Science, PubMed, and Scopus, using a carefully designed set of keywords such as 'Chinese tourists', 'Malaysia tourism', 'visit intention', and 'semi-structured interviews'. Special attention was given to studies about Chinese tourists' plans to visit or those who have visited Malaysia and related research on visit intentions, behaviors, or destination perceptions.

After the initial search, about 173 potentially relevant articles were identified. Each article's title, abstract, and keywords were carefully reviewed, and a first round of filtering was performed based on clear inclusion and exclusion criteria. Inclusion criteria included studies on the visit intentions of Chinese tourists regarding Malaysia using quantitative and qualitative methods such as semi-structured interviews or thematic analysis, as well as those published in peer-reviewed journals. Exclusion criteria included studies not directly related to the research theme, those with unclear methodologies, and non-English articles (unless an English translation was provided). After filtering, 15 articles were included for detailed analysis [13,17,21,22,25–35].

To ensure the systematic and comprehensive nature of the literature review, this study also developed a detailed data extraction strategy, extracting key information from each included article, such as study objectives, methods, and main findings, which were compiled into a comprehensive data table. Additionally, a quality assessment was conducted on the included articles to ensure the reliability and validity of the research findings. Detailed information about the literature selection process, inclusion and exclusion criteria, data extraction, and quality assessment will be provided in Appendices A and B through flowcharts and tables, allowing readers to have a clear understanding of and evaluate the methodological decisions of this study.

By following the PRISMA protocol, this literature review ensures clarity in research design and transparency in decision-making, providing a solid theoretical basis for exploring and understanding the key factors influencing Chinese tourists' intentions to visit Malaysia.

### 2.1. Visit Intention

Visit intention is defined as the measure of an individual's or group's psychological inclination and intention to travel to a specific location or participate in particular activities [23]. Scholars broadly agree that there is a significant interplay between tourists' visit intentions and their chosen tourism destinations [21,24,36]. For tourists, visit intention represents an internal driving force, formed through a comprehensive evaluation of their perceptions, emotional responses, expectations, and other factors regarding the destination, ultimately determining their choice to visit [23,37]. For destinations, visit intention not only enhances tourism revenue and employment [22] but also promotes economic and sociocultural development [23], while supporting environmental protection and sustainable tourism initiatives [21]. In the post-pandemic recovery phase, bolstering visit intention is

crucial for the restoration of tourist flows, as steady increases in visitor numbers positively impact the tourism-related economic development of the destination countries [23,38].

*2.2. Factors Influencing Visit Intention*

The dimensions of image, attractiveness, and competitiveness are fundamental in evaluating tourists' intentions to visit a location. Research by [16,19,21] highlights the significant reference value of these dimensions for understanding and predicting tourists' choice behavior. Ahmad et al., [23] emphasize the critical role of destination image in determining tourists' visit intentions, especially in the post-pandemic context. Similarly, Saad et al., [16] regard tourism attractiveness as a core indicator for assessing tourists' preferences and satisfaction, directly influencing their visit intentions. Moreover, enhancing destination competitiveness is considered an effective strategy to boost visit intention, increasing tourism demand, supported by the findings of [17,19]. Importantly, Crouch and Ritchie [14] note that the dimensions of image, attractiveness, and competitiveness not only have their independent effects on visit intention but also interact synergistically, collectively forming the complex structure of tourists' decision-making processes.

Emotion, as a subjective factor, significantly influences tourists' visit intentions. Santos et al., [39] revealed that emotions are closely linked to satisfaction, willingness to recommend, and overall perception of the destination image, which, in turn, affect changes in tourists' visit intentions. Negative emotional responses from tourists, such as displeasure and disappointment, can markedly hinder their visit intentions and recommendations [40,41]. Prayag et al., [42] found that while positive emotions like happiness, love, and surprise are equally important, the impact of negative emotions tends to be more enduring and pronounced. Research by Yu et al., [33] further disclosed that the correlation between tourists' negative emotions and insufficient visit intentions is modulated through the perceived risks associated with the travel destination. Similarly, Chua et al., [43] found that negative emotions could affect individuals' psychological well-being and attitudes towards tourism by heightening the perception of health risks, ultimately leading to a decline in visit intentions. The underlying reason is that such negative emotions could potentially damage the destination's image [40]. Chen et al., [36] obtained similar findings, suggesting that negative factors, such as high perceived risks, could diminish the positive influence of positive emotions on visit intention. Scholars exploring tourists' visit intentions through comments on social media [31] discovered that negative emotions expressed online could weaken other potential tourists' visit intentions.

*2.3. The Impacts of Negative News and Fake News on Tourists' Emotions and Visit Intentions*

The influence of negative news on tourists' emotions has been affirmed. Negative emotions are not solely dependent on the authenticity of the news but also on tourists' personal experiences, emotional states, and interpretations of information [40]. Genuine negative news, such as significant epidemics [23] or poor security conditions [44], can directly impact tourists' sense of safety and travel decisions, leading to concerns and fear regarding specific destinations [44]. Simultaneously, exaggerated or misleading negative publicity can provoke disproportionate panic or concern, even when actual risks are low [45]. The manner and tone of media reports, as well as the channels through which information is disseminated, can influence public emotional responses and risk perceptions [32]. Soroka and McAdams [46] revealed a tendency to give more attention and weight to negative information, meaning that negative news is more likely to capture attention and elicit emotional responses compared to positive or neutral news [40]. Therefore, even exaggerated negative news can leave a lasting impression on tourists, affecting their emotions and behaviors [47], potentially due to the media's portrayal of dangers and crises, amplifying risk perception and, in turn, impacting tourists' emotional reactions [47].

Fake news significantly affects the decision-making, expectations, and behaviors of tourism consumers through its dissemination via social media and news articles, distorting perceptions of tourism destinations, accommodation units, or attractions [48]. Research

indicates that fake news, facilitated by the wide reach of social media, exploits human reactivity to emotional content, particularly negative emotions, to enhance its impact and speed of dissemination [49]. Compared to genuine news, fake news headlines are more emotionally negative, and the text of fake news exhibits specific negative emotions such as disgust and anger, with less expression of positive emotions [50]. Furthermore, the degree of trust people place in fake news is also influenced by their emotional processing styles, with those who rely on emotion more likely to believe in fake news [51].

Fake news has had a profound impact on the visit intentions of potential tourists. Such misinformation not only distorts consumers' perceptions of destinations, accommodation, cruises, and tourist attractions but may also alter their travel decisions [48]. Spread through social media and news articles, fake news leverages emotional reactions to enhance its dissemination effectiveness, with the stimulation of negative emotions being particularly prominent [49]. Research indicates that fake news, by triggering strong emotional reactions, especially negative emotions, affects people's cognitive processes and, therefore, influences their travel choices [51]. Furthermore, the high expression of negative emotions in fake news not only attracts more attention but also makes individuals more likely to share this information with others due to the emotional responses it evokes [50]. This acceleration of information spread through emotional responses further amplifies the impact of fake news on people's visit intentions.

Fake news refers to intentionally fabricated information lacking a factual basis, often designed to mislead the public or secure undue benefits for individuals or organizations [48]. In contrast, negative news may contain information that is unfavorable to someone or something but is based on facts and constitutes a true report of events [50]. Although negative news can provoke strong emotional reactions from the public, its dissemination and acceptance are constrained by the accuracy and credibility of the facts [40], which marks a major difference from fake news. Moreover, the content of fake news is usually entirely concocted or based on some real events but heavily distorted and exaggerated [48]. Negative news, while perhaps focusing on the negative aspects of an event, still adheres to the basic principles of journalism, namely accuracy and fairness [44]. Therefore, distinguishing between fake news and negative news involves not only the truthfulness of the content but also the intent and manner of reporting. Thus, the widely circulated news on Chinese social media that might negatively affect the intention to visit Malaysia should not be classified as fake news but as negative news.

The study of the impact of negative news on visit intentions presents a complex scenario, with academic debate over its real effect on destination choice. Some scholars [41,44,52] believe that negative news can directly reduce visit intentions by eliciting negative emotions in tourists, a trend also observed by [13], especially during the COVID-19 period. Other researchers [33,36,42] argue that negative news indirectly influences tourist choices by damaging the destination's image, reducing its competitiveness, and ultimately leading to a decline in visit intentions.

In contrast, other researchers offer a differing viewpoint, suggesting that the actual impact of negative emotions on behavioral intentions must consider how individuals process such emotions and information [30]. Furthermore, positive interactions on social media can mitigate the impact of negative news [53]. Nawijn and Biran [40] extended this argument, proposing that, in some cases, negative emotions could have a positive effect on the tourism experience, challenging the traditional view that negative emotions always negatively influence tourism decisions. Yu et al., [33] also found that environmental risks have a more critical impact than the influence of negative news on visit intention. The debate possibly arises because the factors influencing the impact of negative news on visit intention are diverse and vary across different contexts, leading to non-uniform effects.

Although sentiment analysis based on UGC can offer deep insights into tourists' emotional responses to negative news, this research relies on extensive foundational data, and sudden events often fail to generate sufficient commentary on social media in a short time frame [54,55]. Additionally, when evaluating travel destinations, Chinese tourists tend

to use indirect feedback methods, even using anonymous social media platforms [56]. A possible reason is the influence of a collectivist and harmony-oriented culture, where emotional expression is typically more restrained and conservative [57]. Therefore, obtaining ideal results through such methods when investigating whether negative news generates negative emotions in Chinese tourists sufficient to change visit intentions can be challenging.

### 2.4. Factors Influencing the Intent to Visit Malaysia

In the exploration of tourism intentions towards Malaysia, the existing literature primarily relies on survey methodologies for data collection, with Structural Equation Modeling (SEM) serving as the mainstream statistical tool for data analysis (e.g., [15–17,20]). Scholars have focused on single dimensions such as destination image (e.g., [15,21,22]), attractiveness (e.g., [16]), or competitiveness (e.g., [17–20]) to examine tourists' intentions to visit Malaysia from various perspectives. These perspectives include urban tourism [18], rural tourism [20], and coastal tourism [19], as well as tourism projects such as sports tourism [21] and cultural tourism [22]. However, focusing on a singular dimension may not fully reflect the combined effect of multiple factors on visit intentions, and the accuracy of studies examining Malaysia's overall tourist visit intentions from a macro perspective requires further validation [58]. This limitation often restricts a deeper understanding of tourists' decision-making behaviors [59]. To the best of our knowledge, only [37] have attempted a comprehensive study incorporating the dimensions of tourism image, attractiveness, and competitiveness, yet this research also overlooked the social influences and cultural differences among tourists [60].

In terms of research subjects, scholars have limited their sample collection to tourists from Southeast Asia and the Middle East [17], Europe and Oceania [16], or domestic residents [20], with relatively little focus on Chinese tourists. Moreover, although surveys are widely used for data collection, they have limitations in exploring tourists' deeper emotions, as highlighted by refs. [12,61]. The prevalent use of cross-sectional designs further constrains researchers' ability to analyze and understand how tourist behavior changes over time, which could lead to biases in examining whether negative news impacts tourism visit intentions.

An increasing number of researchers are exploring these issues by analyzing images or comments posted by tourists and destination operators on international social media platforms such as Facebook, Agoda, TripAdvisor, and Flickr [62–64]. However, Azazi and Shaed [65] argue that studies on how social media influences tourists' intentions to visit are still limited, as recent research has focused on several key areas: Firstly, developing destination recommendation systems to foster a positive tourism image has garnered attention [63]. Secondly, researchers [62,66] have deepened understanding of visit intentions by analyzing points of interest to construct tourist behavior analysis models and visitor flow prediction models. Thirdly, new research directions include analyzing online reviews to reveal consumer behaviors and proposing models for assessing the competitive advantages and disadvantages of tourism products based on UGC [29,64]. These studies attempt to enhance understanding of visit intentions through various approaches, but none have specifically addressed the main factors influencing the visit intentions regarding Malaysia.

## 3. Methodology

Interview-based research methods, known for their depth and flexibility, have been widely recognized for exploring the impacts of sudden events on behavioral intentions [67]. This approach is particularly advantageous when analyzing complex phenomena such as the effect of negative news on tourism intentions, as it effectively captures the emotional responses of tourists [27,68].

In this study, we used semi-structured interviews to collect data and utilized thematic analysis to process and analyze the data, aiming to identify and understand the complex factors influencing tourists' intentions to visit. Through open-ended questions, we delved into the perspectives of Chinese tourists and tourism practitioners while guiding respon-

dents to explore potential barriers to tourism. The interview questions and guide can be found in Appendix A.1, and the reference table for the interview questions and guide is available in Appendix A.2.

In our research, we adopted the methodological strategy of data triangulation to enhance the credibility and validity of our findings [69,70]. Specifically, we achieved this by collecting and analyzing data from three distinct groups: tourists who visited Malaysia in 2023, those who planned to visit but ultimately canceled their trips, and local tourism operators and managers serving Chinese tourists. This method of collecting and analyzing data from multiple sources reflects the core principle of data triangulation, which is to strengthen the reliability of research findings by utilizing different data sources [71].

Furthermore, by integrating perspectives from different stakeholders, we aimed to comprehensively uncover the multifaceted factors influencing Chinese tourists' intentions to visit Malaysia. Conducting thematic analysis on these diverse sources allowed us to identify and validate common themes and insights in this study, thereby enhancing the reliability of the results. Our methodological approach is supported by the existing literature, which suggests that data triangulation contributes to a more detailed and comprehensive understanding during the research process [69,70].

*3.1. Data Collection*

In January 2024, we engaged the services of China International Travel Service Limited to identify appropriate participants who were briefed on the goals of the research, the importance of diversity, and the principles of theoretical saturation, as well as the objectivity and adequacy of sample size and duration, following the recommendations provided by [72,73]. Our methodology involved segregating participants into two distinct groups for semi-structured interviews: a Tourist Group (Group A), consisting of 30 individuals, and a Host Group (Group B), also comprising 30 individuals. The duration of these interviews ranged from 31 to 36 min, with an average length of 33 min, thereby not only augmenting the data pool but also bolstering the trustworthiness of our findings [74].

Throughout the sampling process, we engaged in ongoing data comparison, utilizing the gathered information to inform the selection of further participants and ensure that each addition to the pool offered a meaningful contribution to the theoretical framework. Adherence to the ethical standards outlined in the Declaration of Helsinki was strict; we obtained consent for the recordings prior to conducting interviews and ensured that participants had the opportunity to review the recordings afterward to validate their content, thus safeguarding their rights. The provision of data was entirely voluntary, with participants being encouraged to communicate their level of comfort at any stage during the interview sequence. To ensure privacy, all collected data were encrypted and are slated for destruction following publication.

Drawing on past experiences [75], we selected respondents for Group A, including tourists who visited Malaysia in 2023 and those who planned to visit but were unable to do so. These two categories of respondents were chosen in equal proportions. This recruitment strategy was designed to minimize reliance on retrospective memory while ensuring that both categories of respondents could equally contribute information, providing a more comprehensive research perspective. The selection of this specific group also aligns with the requirements of thematic analysis methodology, which emphasizes the importance of focusing on current and contextually relevant experiences [73].

For Group B, we selected Malaysian practitioners and managers who have served Chinese tourists. This strategy aimed to delve deeper into these professionals' understanding of the potential needs of Chinese tourists. To address potential sample biases and ensure the achievement of research objectives, we conducted a mid-term assessment based on the recommendations of [75], checking progress and preliminary data to ensure alignment in direction and strategy among all members.

The interviews were conducted in Shanghai and Suzhou, China, and Kuala Lumpur and Penang, Malaysia. To ensure the richness and diversity of data, we initially conducted a

round of interviews with Group A respondents in Shanghai. As a pioneering city in China's open tourism policy with a large population of tourists with outbound travel experience, Shanghai provided an ideal location to gather preliminary insights into intentions to visit Malaysia. Following the initial coding and analysis of the Shanghai interview data, several themes were identified. To ensure depth and breadth in our research, we selected Suzhou as the second interview location. The geographical and cultural proximity of Suzhou to Shanghai allowed us to explore and compare the perspectives of travelers from both cities while minimizing the impact of regional differences. Interviews with Group B began in Kuala Lumpur, followed by Penang. The choice of Kuala Lumpur and Penang as research cities for Group B was due to the substantial number of tourism managers and practitioners in these cities, who have accumulated extensive experience in hosting Chinese tourists.

*3.2. Data Processing*

Following the completion of data collection, we employed thematic analysis to further process and analyze the data. In line with the recommendations proposed by [76], we initially conducted a preliminary coding of the data to identify key concepts and meanings. Subsequently, after accumulating a sufficient volume of coded information, we categorized these elements to facilitate the identification of themes. Upon the preliminary identification of themes, another author reviewed these themes and their relationship with the data. This step aimed to ensure that the identified themes comprehensively and accurately reflected the core elements of the data and revealed their underlying meanings. Throughout the review process, no themes were excluded; instead, the scope of themes was expanded to accommodate a range of experiential elements reported within the same theme.

In the reviewing themes phase, we augmented Group A with five members and Group B with four members to ensure theme saturation. When further interviews failed to reveal new information or themes, this indicated data saturation, affirming the research design's effectiveness, as validated by [77]. We then merged the temporary themes from both groups. This approach facilitated a comprehensive understanding of the factors affecting tourists' visit intentions, incorporating both common points of interest and unique perspectives, a method supported by previous researchers [73]. Furthermore, due to the discovery of overlaps and correlations between themes, merging similar themes became a necessary step for streamlining the analysis and precisely identifying key factors [76].

During the defining and naming themes process, we sought to uncover the deeper logic between themes and further merged them, a practice also endorsed by [76]. Subsequently, we compared the derived themes with the relevant literature. The credibility of our research was established when continuous comparison with literature yielded no new concepts or themes [73].

Table 2 illustrates the process from identifying to finally naming a theme, "Language Proficiency and Multilingual Services to Attract Chinese Tourists". The interviews related to this theme primarily focused on how language barriers diminish the quality of travel experiences. We extracted and coded data from interviews, categorizing them according to their characteristics and attributes based on relevance. Logically related codes were then compiled to extract temporary themes. During the searching for themes phase, this potential theme yielded 11 temporary themes and 27 supporting codes. Subsequently, we merged the temporary themes from both groups and redefined them, resulting in the consolidation of 11 temporary themes into 5 integrated themes. Additional interviews did not yield new data, leading to the theme being deemed saturated. This study then moved into the defining and naming theme phase, during which we discovered deeper connections between the 5 integrated themes. Following previous recommendations [76,78], we classified these integrated themes into a broader theme that emphasized the significant impacts of language proficiency and multilingual services on Chinese tourists' travel experiences and destination image, as well as their key role in enhancing visit intentions. A literature review confirmed that this theme aligns with existing research findings [79], which suggest that focusing on language proficiency enhancement and providing multilingual services can effectively boost Chinese tourists' visit intentions regarding Malaysia.

**Table 2.** Example of the thematic process.

| Perspective | Group | Data (Typical) | Coding | Searching for Themes | Reviewing Themes | Defining and Naming Theme |
|---|---|---|---|---|---|---|
| Image | A | Respondent C in Shanghai: "Language barriers sometimes make me feel inconvenient and confused. This communication barrier somewhat affects my view of the country's tourism image. | Language barriers leading to inconvenience and confusion | Temp theme 1: Negative impact of language barriers on travel experience; Temp theme 2: Importance of multilingual services in enhancing tourism image. | Integrated theme 1: The comprehensive impacts of language services on tavel experience and destination image; Integrated theme 2: The critical role of language proficiency in enhancing tourism competitiveness | Final theme: Language proficiency and multilingual services to attract Chinese tourists |
| | | Respondent R in Shanghai: Language is not only a tool for communication but also significantly affects my connection with the destination, thereby influencing my overall view of the country's tourism image". | | | | |
| | | Respondent F in Shanghai: "If scenic spots and tourism services could provide more comprehensive multilingual support, such as multilingual signs and guided services, I believe Malaysia's tourism image would be much more friendly". | Multilingual support and improvement of tourism image | | | |
| | | Respondent M in Suzhou: "Lack of effective language communication when exploring traditional markets or participating in local activities makes me feel alienated, affecting my overall impression of the country as a tourism destination". | Language barriers and social alienation | | | |
| | | Respondent O in Suzhou: "When I tried to try various local snacks on the snack street, I found myself unable to communicate effectively with the vendors". | | | | |
| | | Respondent Q in Shanghai: "I can't understand the conversations and specific meanings of activities around me. This language barrier makes me feel like an outsider". | | | | |
| | B | Respondent K in Penang: "We are striving to improve our staff's Chinese communication skills to enhance the experience of Chinese tourists. I believe this will not only enhance Malaysia's tourism image but also make us more competitive in the tourism market". | Improving Chinese communication skills to enhance tourism image and competitiveness | Temp theme 3: Enhancing tourism competitiveness through improved Chinese communication skills; Temp theme 4: Language ability is related to the improvement of tourism experience and image. | | |
| | | Respondent N in Penang: "As a Chinese tour guide, when I can explain in Mandarin, tourists' reactions are much warmer, which is very helpful for enhancing Malaysia as a tourism destination". | Using Mandarin to enhance travel experience and destination image | | | |
| | | Respondent O in Penang: "They are very satisfied with being able to understand Malaysian culture and attractions in a language they are familiar with". | | | | |
| | | Respondent K in Penang: "Mastering Mandarin is crucial to enhancing Malaysia's tourism experience and destination image". | | | | |

**Table 2.** *Cont.*

| Perspective | Group | Data (Typical) | Coding | Searching for Themes | Reviewing Themes | Defining and Naming Theme |
|---|---|---|---|---|---|---|
| Attractiveness | A | Respondent K in Suzhou: "Not being able to fully understand the stories behind local festivals or food reduced my travel enjoyment. If more language assistance were provided, such as multilingual menus or cultural event explanations, Malaysia's tourism attractiveness would be more evident". | The need for language support to enhance travel experience and attractiveness. | Temp theme 5: The key role of language support in enhancing travel experiences; Temp theme 6: The impact of language barriers on in-depth understanding of and experiencing local culture. | Integrated theme 3: The comprehensive role of language services in enhancing travel experiences and cultural understanding; Integrated theme 4: The dual role of language barriers and services in target market engagement | Final theme: Language proficiency and multilingual services to attract Chinese tourists |
| | | Respondent A in Shanghai: "I often encounter menus only in Malay, which makes it difficult for me to understand each dish's features and the cultural stories behind them". | | | | |
| | | Respondent H in Shanghai: "I cannot communicate deeply with locals and fully experience and understand the local lifestyle and traditions". | Language barriers limit the experience of local culture. | | | |
| | | Respondent L in Suzhou: "When trying to communicate with local Malays or Indians, I found the language differences between us made it difficult to deeply understand their unique living habits and cultural traditions". | | | | |
| | | Respondent G in Shanghai: "Due to language barriers, I find myself unable to communicate effectively and have a hard time truly understanding their stories and traditional knowledge". | | | | |
| | B | Respondent B in Kuala Lumpur: "Providing Chinese services, such as Chinese travel brochures and signs, is crucial for attracting more Chinese tourists". | The importance of Chinese-language services in attracting Chinese tourists. | Temp theme 7: The core role of Chinese services in attracting Chinese tourists; Temp theme 8: The crucial role of Chinese-speaking guides in promoting cultural understanding and enhancing the travel experience. | | |
| | | Respondent D in Kuala Lumpur: "I find Chinese signage and manuals extremely important for attracting Chinese tourists. Since providing these services, tourists from China have significantly increased, and their travel experiences have improved". | | | | |
| | | Respondent E in Kuala Lumpur: "After introducing Chinese services, the satisfaction of Chinese tourists has significantly improved. These services make them feel more welcomed and have also helped us attract more Chinese tourists". | | | | |
| | | Respondent G in Kuala Lumpur: "By providing Chinese guide services, I can communicate better with Chinese tourists and explain Malaysia's history, culture, and customs. I believe this cultural exchange not only enriches the tourists' travel experience but also enhances the attractiveness of Malaysia's tourism". | Chinese guide services enhance cultural exchange and tourism attractiveness. | | | |
| | | Respondent D in Kuala Lumpur: "Explaining our history and culture in Chinese not only improves the experience of tourists but also increases the attractiveness of our tourism". | | | | |
| | | Respondent C in Kuala Lumpur: "Through Chinese services, I have deepened connections with Chinese tourists. Explaining culture and customs not only enriches their experience but also improves our tourism appeal". | | | | |

**Table 2.** *Cont.*

| Perspective | Group | Data (Typical) | Coding | Searching for Themes | Reviewing Themes | Defining and Naming Theme |
|---|---|---|---|---|---|---|
| Competitiveness | A | Respondent L in Suzhou: "Providing services in more languages at airports, hotels, and tourist spots will undoubtedly enhance their competitiveness in the international tourism market". | Multilingual services enhance international tourism market competitiveness | Temp theme 9: Key role of multilingual services in enhancing international tourism competitiveness; | Integrated theme 5: The central role of multilingual services and professional language skills in enhancing the international competitiveness of the tourism industry | Final theme: Language proficiency and multilingual services to attract Chinese tourists |
| | B | Respondent M in Penang: "Being able to provide quality Chinese-language services is key to enhancing competitiveness". | Quality Chinese services enhance competitiveness | Temp theme 10: The importance of Chinese services in enhancing the competitiveness of the tourism industry; Temp theme 11: The role of fluent Chinese-speaking guides in enhancing personal and industry competitiveness. | | |
| | | Respondent F in Kuala Lumpur: "I firmly believe that providing quality Chinese-language services is key to attracting more Chinese tourists. These services not only make tourists feel more comfortable but also significantly enhance our competitive edge". | | | | |
| | | Respondent A in Kuala Lumpur: "This kind of (Chinese) service makes the tourist experience more personalized and satisfactory". | | | | |
| | | Respondent H in Penang: "Guides who can speak fluent Chinese are more popular. This not only enhances our competitiveness as guides but also improves the overall competitiveness of Malaysia's tourism industry". | Fluent Chinese-speaking guides enhance both personal and industry competitiveness | | | |
| | | Respondent K in Kuala Lumpur: "Providing services in Chinese enhances the entire Malaysian tourism industry's appeal to Chinese tourists, strengthening our industry's competitiveness". | | | | |

Source: the author's own data analysis.

Refs. [75,80] both emphasize the importance of reflexivity as a critical methodological tool in the practice of thematic analysis, ensuring depth and richness in research. Compared to reliance on software tools for mechanical coding, active engagement and reflexive thinking play a more pivotal role in uncovering the deep meanings within data. Consequently, this study utilized NVivo 12 solely for transcribing collected recordings into text, with the entire coding process being conducted manually.

## 4. Results

As shown in Table 3, we interviewed 69 participants, resulting in approximately 641,100 words of transcribed text. During the coding phase, we identified 159 codes. As indicated in Appendix B, in the searching for themes stage, 17 temporary themes were identified by each group. During the reviewing themes stage, by categorizing and merging similar views from both groups, we derived 6, 7, and 6 integrated themes in the dimensions of tourism image, attractiveness, and competitiveness, respectively.

**Table 3.** An overview of the interviews.

| Groups | Location | Number of Interviewees | Total Valid Recordings (h/min) | Characters Converted from Recordings (10 K) |
|---|---|---|---|---|
| Tourist Group (Group A) | Shanghai | 15 + 5 | 10 h 21′ | 17.04 |
| | Suzhou | 15 | 8 h 13′ | 13.55 |
| Host Group (Group B) | Kuala Lumpur | 15 + 4 | 11 h 18′ | 18.64 |
| | Penang | 15 | 9 h 1′ | 14.88 |

The previous literature (e.g., [75,77]) often involves authors defining and naming identified themes. Our study adopts the approach of [35], inviting two trained coders with extensive knowledge in the field of destination visit intention to independently define and name the themes. The coders then discussed their findings and ultimately generated a reconciled theme name. This Collaborative Coding Technique (CCT) provides diverse perspectives, introduces new information, and suggests novel interpretations, thereby enhancing the depth and breadth of the research and ensuring the accuracy and applicability of theme definition and naming.

Following this, based on the complete interview data, we constructed a vector containing all interview IDs, serving as the total population set for sampling. As suggested by prior research [35], two additional researchers re-verified the interview materials to ensure the reliability of the naming process. To ensure the comprehensiveness and diversity of the sample, we planned to randomly select two to three interview materials from each location for in-depth analysis. This process was implemented using a sampling function, aimed at ensuring a fair and random selection process from the interview dataset. The sampling was executed without replacement to avoid the repetition of the same interview material, thus ensuring the uniqueness and representativeness of the sample. The specific sampling function is shown below:

$$S(X, m, False, P) \rightarrow \{xi1, xi2, \ldots, xim\} \tag{1}$$

$X$ represents the total population set to be sampled, which is a vector composed of all interview IDs.

$m$ denotes the number of samples required, set at two to three per location based on specific circumstances.

*False* specifies the use of a sampling method without replacement to prevent the repetition of samples.

*P* represents the vector of extraction probabilities assigned to each element, where each interview has an equal probability of being selected, indicating that weighted sampling was not employed in this research.

As shown in Table 4, the agreement levels between CCT and researcher A, CCT and researcher B, and researcher A and researcher B were 0.6102, 0.6098, and 0.618, respectively. These figures indicate a substantial level of consistency [81], suggesting that the naming outcomes obtained through CCT are reliably consistent to a significant extent [81].

**Table 4.** Results of inter-researcher reliability tests.

| Theme | CCT-Researcher A | CCT-Researcher B | Researcher A-Researcher B |
|---|---|---|---|
| | Kappa (Interpretation) | Kappa (Interpretation) | Kappa (Interpretation) |
| Theme 1: language proficiency and multilingual services to attract Chinese tourists | 0.710(Substantial) | 0.760 (Substantial) | 0.850 (Almost perfect) |
| Theme 2: promoting local culture to attract Chinese tourists | 0.707 (Substantial) | 0.758 (Substantial) | 0.820 (Almost perfect) |
| Theme 3: lack of competitive advantage in natural resources | 0.658 (Substantial) | 0.712 (Substantial) | 0.666 (Substantial) |
| Theme 4: marketing and product innovation for the Chinese market | 0.638 (Substantial) | 0.669 (Substantial) | 0.627 (Substantial) |
| Theme 5: service excellence in accommodation and catering | 0.607 (Substantial) | 0.364 (Fair) | 0.461 (Moderate) |
| Theme 6: maintaining satisfaction in healthcare and hygiene | 0.559 (Moderate) | 0.556 (Moderate) | 0.60 (Moderate) |
| Theme 7: In-depth religious experience for Chinese tourists | 0.453 (Fair) | 0.391 (Fair) | 0.327 (Fair) |
| Theme 8: Sustaining satisfaction with infrastructure and service quality | 0.870 (Almost perfect) | 0.962 (Almost perfect) | 0.823 (Almost perfect) |
| Theme 9: Tackling regional disparities | 0.378 (Fair) | 0.397 (Fair) | 0.692 (Substantial) |
| Theme 10: Chinese tourists' preferences: traditional experiences over emerging tourism | 0.522 (Moderate) | 0.529 (Moderate) | 0.319 (Fair) |

Kappa: [81] Kappa coefficient.

Our study identified 10 final themes, which are key areas related to enhancing Chinese tourists' visit intentions regarding Malaysia. These themes include language proficiency and multilingual services to attract Chinese tourists, promoting local culture, a lack of competitive advantage in natural resources, marketing and product innovation tailored for the Chinese market, service excellence in accommodation and catering, maintaining satisfaction in healthcare and hygiene, providing in-depth religious experiences for Chinese tourists, sustaining satisfaction with infrastructure and service quality, tackling regional disparities, and Chinese tourists' preferences for traditional experiences over emerging tourism trends.

Ref. [76] emphasize that an effective approach for conducting thematic analysis involves integrating the literature review and results validation stages. As shown in Table 5, we compared the outcomes of our thematic analysis with the findings from five prior studies [16,19,64,79,82] concerning Malaysia's visit intentions. According to [78], the findings of a thematic analysis should correspond with existing research. Our comparison confirmed six previously identified themes, validating the effectiveness of our research method and demonstrating its ability to confirm and verify established knowledge. Additionally, due to its methodological clarity and analytical flexibility [73], our research identified four new themes not previously mentioned in the literature, thereby expanding the understanding of this field.

**Table 5.** Comparison between the themes obtained by the Thematic Analysis method and those obtained by other methods.

| This Study | [79] | [82] | [16] | [19] | [64] |
|---|---|---|---|---|---|
| language proficiency and multilingual services to attract Chinese tourists | Extensive challenges of language barriers in international environments | Chinese tourists in Malaysia still need more language-related care | | | |
| promoting local culture to attract Chinese tourists | N/A | N/A | N/A | N/A | N/A |
| lack of competitive advantage in natural resources | N/A | N/A | N/A | N/A | N/A |
| marketing and product innovation for the Chinese market | | Focused on specific tourism product innovation and digital marketing strategies for the Chinese market | | | |
| service excellence in accommodation and catering | | | | | The hotel services in Malaysia are a guarantee to enhance tourists' travel intentions |
| maintaining satisfaction in healthcare and hygiene | | The importance of medical, health, and hygiene conditions in Chinese tourists' visits to Malaysia | | | |
| in-depth religious experience for Chinese tourists | N/A | N/A | N/A | N/A | N/A |
| sustaining satisfaction with infrastructure and service quality | | | | Infrastructure is key to enhancing satisfaction during travel in Malaysia | |
| tackling regional disparities | | | Unleashing the potential of inbound tourism in Malaysia by addressing service challenges and regional disparities to enhance the travel experience | | |
| Chinese tourists' preferences: traditional experiences over emerging tourism | N/A | N/A | N/A | N/A | N/A |

## 5. Discussion

Despite scholars such as Robertson et al., [52] suggesting that negative news can provoke adverse emotional reactions among tourists, research focused on the Malaysian market by [7,10] has underscored a connection between negative reporting, mystery films, and negative emotions, which, in turn, affects tourists' visit intentions. However, our research reveals that these negative emotions are not the decisive factors leading to a lack of visit intention. Our findings echo those of [83], indicating that Chinese tourists are not influenced by external instabilities and negative news.

### 5.1. Basic Theoretical Discussion

Our study identified ten primary reasons for the decline in tourist flows to Malaysia, with six of these reasons having been addressed in previous studies. Our study provides a more in-depth analysis, building on their foundation.

Language proficiency and multilingual services to attract Chinese tourists. This theme highlights that Chinese tourists' travel experiences are hindered by language barriers, affecting their understanding of culture, history, and even normal tourism activities. A typical discussion stated "Visiting museums, we found no Chinese introductions, not even Chinese signage, forcing us to guess our way through the exhibits". Another example is "At a Mamak stall, the language barrier led to ordering the wrong meal as they only had menus in Malay. Despite the food being delicious, it felt like something was missing". This may be attributed to the lack of sufficient multilingual services at tourist destinations, causing difficulties for Chinese tourists in understanding culture, food, and history. This finding supports prior research showing that multilingual services, especially Chinese-language support in Malaysia, improve Chinese tourists' experiences and satisfaction by facilitating cultural understanding and engagement. Specifically, these studies underscore the importance of providing Chinese-language support for Chinese tourists in Malaysia, particularly in facilitating cultural understanding and engagement. [1,79,82]. However, previous studies have not deeply explored the specific impact of language barriers on tourists' visit intentions.

Marketing and product innovation for the Chinese market. This theme reveals that Malaysia's strategy for marketing tourism products to Chinese visitors falls short in incorporating essential business insights, underscoring the imperative for adopting marketing strategies that are both more strategic and innovative. An example statement is "the current promotional methods fail to effectively capture the attention of Chinese tourists". This ineffectiveness may stem from the insufficient presence of Destination Management Organizations (DMOs) on Chinese social media platforms, despite young Chinese tourists significantly valuing social media promotions and recommendations in their travel planning, as noted by ref. [84]. Furthermore, research indicates a misalignment between tourism products and marketing messages and the specific needs and expectations of Chinese tourists, revealing a gap in the understanding of the target market. An illustrative remark is "Through detailed research into the Chinese market, we've discovered that personalizing tourism products and marketing strategies significantly enhances Chinese tourists' satisfaction and willingness to participate". This insight is supported by the works of [34,36,82], which emphasize the importance of developing targeted tourism product innovations and digital marketing strategies for Chinese tourists. Utilizing emerging technologies and social platforms can not only more effectively attract Chinese tourists but also meet their desires for novel experiences and personalized services.

Service excellence in accommodation and catering. This topic reflects the general satisfaction of Chinese tourists with the hospitality and catering services in Malaysia. An example statement is "A hotel in Kuala Lumpur not only provided a comfortable living environment but also offered a variety of Chinese breakfast options, making us feel at home". Another example is "We were assigned a room with a sea view, beautifully decorated to reflect Malaysian cultural characteristics. The resort also offered tailored travel itineraries and personalized services, making our travel experience more unique and

memorable". This aligns with the research by ref. [64], which posits that Malaysian hotel services guarantee an increase in tourists' intentions to travel.

Maintaining satisfaction in healthcare and hygiene. This theme reveals that Chinese tourists express high satisfaction with Malaysia's sanitation and healthcare conditions, deeming them adequate for their health and safety needs. One comment contains the following: "In Kuala Lumpur, we found that the cleanliness and hygiene standards in public places such as shopping centers, restaurants, and hotels were very high, which reassured us". Another example is "When I was accidentally injured in Penang, the swift access to high-quality medical services left a lasting impression on me about Malaysia's healthcare system". This supports ref. [82] finding that healthcare and sanitary conditions are crucial for the satisfaction of Chinese tourists in Malaysia.

Sustaining satisfaction with infrastructure and service quality. This theme indicates that Chinese tourists are satisfied with the infrastructure and service quality in major cities. A typical statement is "We are satisfied with the light rail and subway systems in Kuala Lumpur, as they are not only convenient and efficient but also very clean". Another example is "The widespread availability of free high-speed Wi-Fi in airports and hotels in Malaysia has left a deep impression on us". Our finding is consistent with research claiming that solid infrastructure significantly improves Chinese tourists' satisfaction with Malaysia [19], underlining the importance of good infrastructure for tourist satisfaction and sustainable tourism.

Tackling regional disparities. Our results highlight that Chinese tourists exhibit a certain level of dissatisfaction with the regional differences within Malaysia, especially when comparing major tourist cities with other areas. A typical discussion is as follows "There are differences in infrastructure between East Malaysia and West Malaysia, which affect the overall travel experience of tourists". Another example statement is "Non-tourist cities and islands have significant infrastructure disparities compared to Penang, Malacca, and Kuala Lumpur, making us often miss the convenience and comfort of these big cities". Our findings are consistent with the research by ref. [16], which pointed out that addressing developmental disparities between regions is key to unlocking Malaysia's tourism potential and enhancing the travel experience.

*5.2. Discussion of Key Findings*

Promoting local culture to attract Chinese tourists. This theme reveals that Chinese tourists show a deep interest in experiencing the authentic cultural characteristics and lifestyles of residents. One common sentiment is "We want to visit their farmers' markets, eat in local restaurants, and even have the chance to be their guests". However, travel agencies often do not offer these kinds of experiences. Another viewpoint emphasizes the following: "We are eager to participate in local festivals and workshops, experiencing traditional crafts and cooking firsthand. These experiences let us dive deeper into the local culture than just visiting the usual tourist spots". Ref. [85] provides a theoretical foundation for these observations, suggesting that deep engagement with a destination's native culture can positively affect tourists' perceptions of that destination. However, this view does not fully capture the fact that native culture is not only found in cultural heritage or tourism resources. The day-to-day lives of local communities are filled with culturally rich details that are highly attractive. Chinese tourists are no longer satisfied with traditional sightseeing tours; exquisite tourism products rich in local cultural characteristics are becoming their preferred choice [1]. Our study finds that Malaysia's shortcomings in showcasing its indigenous cultural features could negatively impact its overall tourism image, leading to a lack of desire to visit. This perspective is also supported by [86,87], who discovered that showcasing local cultural features and lifestyles is crucial for attracting tourists and increasing their visit intentions.

Lack of competitive advantage in natural resources. despite numerous scholarly articles (e.g., [18,19,21]) championing Malaysia's unique natural tourism resources as a catalyst for enhancing tourist interest, our study indicated that Chinese tourists perceive a

homogeneity in Malaysia's tourism offerings compared to neighboring countries. A typical statement is "While Malaysia's natural landscapes undeniably possess allure, this appeal does not distinctly set it apart from the tourism resources of adjacent nations in the eyes of visitors". Furthermore, another observation highlights the following: "I noticed that Malaysia's beaches bear a striking resemblance to those offered by Thailand, Vietnam, and Indonesia, rendering it less distinguished among a plethora of choices". These findings support previous studies that claimed that there is a high degree of similarity in the natural tourism resources among Southeast Asian countries [88].

In-depth religious experience for Chinese tourists. Despite ref. [28] arguing that Chinese tourists lack interest in local religious cultures, our research underscores the potential for enhancing Chinese tourists' visit intentions by integrating religious culture into tourism products, though current efforts appear to be insufficient. One common insight is "Even though Chinese tourists are really interested in Islamic culture, it's hard for them to deeply understand and experience it". Another example is "Taking part in activities in mosque is exciting, but it's challenging to fully grasp the meaning behind the observed religious practices, leaving Chinese tourists feeling confused". Feedback also confirms this issue: "Mosques are beautiful, but I don't really get the essence of Islamic culture". Ref. [89] pointed out that cultural differences are the root cause of this understanding barrier. Our analysis reveals a deeper issue, suggesting the problem is not whether Islamic culture itself is unattractive to Chinese tourists but that current tourism promotions for visitors lack specific education and guidance on the differences in religious customs. A typical discussion suggests that "There should be a process of interaction and participation, allowing tourists to not just be onlookers but to actively experience and feel different cultures". Hence, promoting communication and understanding between different cultural backgrounds can significantly increase individuals' visit intentions and engagement with these cultures, a finding also supported by ref. [90]. In fact, as ref. [1] stated, the proportion of Chinese tourists who desire personalized and customized travel experiences is rapidly increasing. Tourism products rich in local religious and cultural features are precisely what these tourists seek, and this will positively impact their visit intentions.

Chinese tourists' preferences: traditional experiences over emerging tourism. A typical discussion contains the following: "City sightseeing in Kuala Lumpur is definitely not to be missed". Another example is "Penang and Malacca are places where Chinese immigrants' heritage in Southeast Asia is preserved". These phenomena might be explained by ref. [25], who noted that the travel patterns of Chinese tourists to Malaysia have not changed much since before the pandemic, with city sightseeing and leisure activities still being their preferred choices. Studies also indicated that despite Malaysia's tourism image-building efforts to introduce new forms of tourism like agro-tourism, medical tourism, or sports tourism to Chinese tourists, respondents did not express having participated in or heard about these new forms of tourism during their visits to Malaysia. A common discussion them is the following: "In serving Chinese tourists, we found they don't show much interest in medical tourism or sports tourism. They seem to prefer traditional tourism experiences, like city sightseeing and cultural exploration". Our findings challenge the assertion, as seen in previous research, that new tourism formats, such as medical tourism, would emerge as major attractions for Chinese tourists post-COVID-19 pandemic and positively transform Malaysia's national tourism image [26]. A possible reason is that China's medical tourism system is gradually developing, and public sentiment is improving, consistent with [91]'s findings that as domestic high-quality medical services have become more widespread, the interest in foreign medical tourism has gradually declined.

## 6. Implications and Limitations

### 6.1. Theoretical Implications

Our work contributes to the literature in the following ways.

Firstly, although questionnaire surveys (e.g., [15–17,20]) and UGC methods (e.g., [62–65]) have provided valuable insights, their limitations become apparent when examining the im-

pact of sudden events related to tourists' deep emotional reactions on their visit intentions. By employing a methodology that combines semi-structured thematic analysis, we have uncovered the main reasons behind the insufficient number of Chinese tourists visiting Malaysia. Additionally, our study identified that negative news on Chinese social media is not the primary factor contributing to the decline in tourism intent. This application of methodology not only pioneers new territories in the study of visitation intention but also offers fresh perspectives on exploring tourists' emotional responses to sudden events.

Secondly, previous limited studies on tourism intention have primarily focused on a single dimension from tourism image, attractiveness, or competitiveness (e.g., [15–17,19]) from specific angles such as coastal tourism, urban tourism, rural tourism [18–20], or particular tourism projects like sports tourism, cultural tourism, etc. [21,22]. Our research framework integrates these three dimensions and assesses Malaysia as a whole, thereby uncovering new findings in a more comprehensive manner.

*6.2. Practical Implications*

This study identifies key factors that hinder the recovery of tourist arrivals, focusing on visitation intentions and providing practical insights for DMOs in Malaysia. Firstly, it is crucial to maintain current advantages in accommodation, catering, and medical services. DMOs should continue to uphold high standards in these areas. Additionally, incorporating cultural integration and personalized experiences can further increase satisfaction and the visit intentions. Secondly, regional disparities in infrastructure development pose a significant issue. These disparities affect the overall travel experience, and DMOs should work towards minimizing these gaps, particularly in infrastructure, to ensure a uniformly high-quality experience across all regions. Thirdly, improving multilingual services and enhancing engagement on Chinese social media are vital. Specifically, offering services in Chinese can significantly enhance the travel experience. Moreover, engaging Chinese tourists through Chinese social media platforms can be an effective marketing strategy, particularly for attracting younger visitors. Fourthly, the promotion of local and religious cultures over natural and emerging tourism products is advisable. Research suggests that an excessive focus on natural landscapes and new tourism offerings, such as sports or medical tourism, may not substantially attract Chinese tourists. Instead, showcasing Malaysia's indigenous and religious cultures could improve cultural understanding and attract interest.

*6.3. Limitations and Future Research*

In this study, while our research provides valuable insights into the topic, it still faces certain limitations. Our findings are primarily applicable under specific conditions where the pandemic is subsiding and visit intentions regarding Malaysia are relatively low, which restricts the applicability and generalizability of our results.

Although this study employed the PRISMA guidelines for the literature review to ensure systematicity and transparency, there are still some limitations to consider. Firstly, the PRISMA guidelines focus on reporting methods for systematic reviews and meta-analyses, whereas our study mainly concentrates on the literature review, which might not fully encompass all research related to the topic. Additionally, due to limitations in the literature and databases, there is a possibility that some key studies may have been overlooked.

In light of the findings from the current study, future research could progress in several directions. Firstly, adopting a mixed-methods research design that combines quantitative approaches (such as surveys or behavioral data analysis) can help to identify and explain potential biases, thereby achieving a more comprehensive understanding. Secondly, cross-cultural comparative studies could illuminate the differences and similarities in visit intentions among tourists from various cultural backgrounds. Additionally, focusing on the impact of digital media and online reputation management on visit intentions, especially during crises such as the post-pandemic period, would provide valuable insights

and guidance for destination managers. Given the limitations identified in the literature review, future research should place greater emphasis on the synthesis of data, integrating qualitative and quantitative methods to obtain more accurate and comprehensive results. Moreover, in-depth studies on specific tourist groups, particularly those sensitive to negative news, could deepen our understanding of their perceptions and behaviors, thus offering destination managers more precise marketing strategies and service solutions. Such research would not only enrich our understanding of visit intentions but also offer practical value to managers of tourism destinations.

Furthermore, in exploring future research directions, a significant and forward-looking topic is the study of Chinese tourists' intention to travel to mainstream Western countries post-pandemic. As the global tourism industry gradually recovers from the impact of COVID-19, understanding the travel preferences, decision-making factors, and destination considerations of Chinese tourists under the new normal becomes particularly crucial.

## 7. Conclusions

Fully understanding the factors that influence tourists' visit intentions is crucial for increasing visitor numbers. This study conducted semi-structured interviews with recent Chinese visitors, those who had planned to visit but were unable to, and local practitioners and managers who have served Chinese tourists. Through thematic analysis, we identified ten main reasons for the insufficient visit intentions regarding Malaysia. We discovered that the lack of intention among Chinese tourists is not primarily due to the negative news frequently reported in Chinese media. Instead, factors such as the absence of language services, ineffective marketing strategies, a lack of engaging local cultural experiences, and insufficient guidance on religious culture are the main deterrents. We also found that Chinese tourists' enthusiasm for urban leisure travel remains strong, and overly promoting emerging tourism products and natural resources does not effectively increase their visit intentions. We further emphasize that DMOs should not overlook the impacts of regional disparities in infrastructure on the visit intentions of Chinese tourists. Our findings aim to promote the sustained development of destinations and the comprehensive recovery of the international tourism industry.

**Author Contributions:** Conceptualization, X.J. and A.E.b.M.; methodology, X.J.; software, A.H.b.A.; validation, X.J. and A.E.b.M. and A.H.b.A.; formal analysis, A.H.b.A.; investigation, A.E.b.M.; resources, A.E.b.M.; data curation, A.H.b.A.; writing—original draft preparation, X.J.; writing—review and editing, X.J.; visualization, X.J.; supervision, A.E.b.M. and A.H.b.A.; project administration, X.J.; funding acquisition, A.E.b.M. All authors have read and agreed to the published version of the manuscript.

**Funding:** This research has received no external funding.

**Institutional Review Board Statement:** The study was conducted in accordance with the Declaration of Helsinki, and approved by the Institutional Review Board of Suzhou SIP Institute of Vocational Technology ethics committee (reference number: [sipivt20240102168]).

**Informed Consent Statement:** Informed consent was obtained from all subjects involved in the study.

**Data Availability Statement:** The data used to support the findings of this study are included within the article.

**Conflicts of Interest:** The authors declare that there are no conflicts of interest regarding the publication of this paper.

## Appendix A

*Appendix A.1. Semi-Structured Interview Questions & Interview Guide*

Semi-Structured Interview Questions
Group A (Tourist Group):

1.   Tourism image:

    1.1.    Describe the tourism image of Malaysia in your mind. What factors have shaped this view?

    1.2.    For tourists who have visited: How do you think the tourism image of Malaysia compares with your actual visit experience?

    1.3.    For tourists choosing not to visit: Are there specific factors or information that influenced your decision not to choose Malaysia as a tourism destination?

2.    Tourism attractiveness:

    2.1.    Before considering a trip to Malaysia, what factors or features do you think could attract you?

    2.2.    For tourists who have visited: In your opinion, which tourist resources or activities offered by Malaysia are most attractive to Chinese tourists?

    2.3.    For tourists choosing not to visit: In your decision-making process, were there elements or features that lacked appeal to you?

3.    Competitiveness:

    3.1.    How do you assess Malaysia's competitiveness in tourism compared to other Southeast Asian countries (such as Thailand, Indonesia)?

    3.2.    For tourists choosing not to visit: Compared to other Southeast Asian countries, in what aspects did Malaysia not attract you?

    Group B (Host Group):

1.    Tourism Image:

    1.1.    How do you evaluate the current tourism image of Malaysia among Chinese tourists? What factors do you think influence this image?

    1.2.    Do you think there is a difference between Malaysia's tourism promotion and the actual tourism experience provided?

2.    Tourism attractiveness:

    2.1.    From your perspective, which tourism resources or services in Malaysia are particularly popular with Chinese tourists?

    2.2.    What unique advantages do you think Malaysia has in attracting Chinese tourists?

3.    Competitiveness:

    3.1.    How do you view Malaysia's competition with other Southeast Asian countries in attracting Chinese tourists?

    3.2.    In what areas do you think the Malaysian tourism industry should focus on to enhance its competitiveness?

    Interview Guide

1.1.    Introduction: Explain the purpose of the interview, confidentiality assurance, the estimated duration of the interview, and consent for recording.

1.2.    Assure the interviewee that their information will be kept confidential and that their participation is voluntary.

1.3.    Specific questions for Group A (Tourist Group) and Group B (Host Group) are designed around three key sections: tourism image, tourist attraction, and competitiveness. When a deeper understanding is needed, questions like: "Can you describe in more detail?" or "Can you share a specific example to illustrate your point?" can be used.

1.4.    At the end of the interview, ask: "Is there anything else you think we should know about your travel experience or opinions?" This question gives the interviewee an open space to share information that may not have been covered.

1.5.    Ensure that all questions and approaches consider cultural differences, avoiding any potential cultural biases or misunderstandings. You might say: "We understand that different cultural backgrounds can influence people's perceptions and experiences, and we welcome hearing your unique perspective based on your cultural background.

*Appendix A.2. Constructs and Indicators*

| Group | | Aspects | Summary of Interview Questions/ Interview Instructions | Reference |
|---|---|---|---|---|
| Semi-structured interview | Group A | Image | Descriptions and factors influencing Malaysia's tourism image | [86] |
| | | Attractiveness | Factors and attractiveness of tourism resources and activities in Malaysia | [36] |
| | | Competitiveness | Comparison of Malaysia's tourism competitiveness with other Southeast Asian countries | [14] |
| | Group B | Image | Image of Malaysia among Chinese tourists and differences in promotion and experience | [26] |
| | | Attractiveness | Popularity of Malaysia's tourism resources services and their unique advantages | [83] |
| | | Competitiveness | Views on Malaysia's competitiveness in attracting Chinese tourists and areas for enhancing competitiveness | [37] |
| Interview Guidelines | | Introduction | Introduction of interview purpose, interview duration, and consent for recording | [92] |
| | | Confidentiality | Assurance of confidentiality and voluntary participation | [93] |
| | | Specific Questions | Open-ended questions designed around the key themes of tourism image, attractiveness, and competitiveness. | [75] |
| | | Conclusion | Asking for any additional thoughts or experiences not covered during the interview | [77] |
| | | Cultural Differences Consideration | Ensuring all questions and strategies consider cultural differences to avoid potential cultural bias or misunderstanding. | [79] |

## Appendix B

**Table A1.** Thematic analysis of Chinese tourists' intentions to visit Malaysia.

| Perspective | Group | Searching for Themes | Reviewing Themes | Defining and Naming Theme |
|---|---|---|---|---|
| Image | A | Temporary Theme A1: Language barriers negatively impact the tourist experience, hindering effective communication and understanding.<br>Temporary Theme A2: The provision of multilingual services is crucial for enhancing the tourism image, facilitating a more inclusive and accessible environment for international visitors.<br>Temporary Theme A3: There is a noticeable ambiguity in the perception of religious and local cultural images, suggesting a need for clearer representation and interpretation to foster a deeper appreciation among tourists.<br>Temporary Theme A4: The diversity of natural and cultural tourism images enriches the tourist experience, offering a wide array of attractions and activities that cater to varied interests.<br>Temporary Theme A5: The portrayal of a harmoniously developed multi-ethnic society significantly contributes to a positive tourism image, showcasing cultural diversity and inclusivity.<br>Temporary Theme A6: Comprehensive tourism infrastructure is essential for a satisfactory visitor experience, encompassing transportation, accommodation, and recreational facilities. | Integrated Theme 1: The Comprehensive Impact of Language Services on Travel Experience and Destination Image.<br>Integrated Theme 2: The Critical Role of Language Proficiency in Enhancing Tourism Competitiveness.<br>Integrated Theme 3: Understanding and Appreciation of Religious and Local Cultural Images in Tourism.<br>Integrated Theme 4: Diversity and Satisfaction in Natural and Cultural Tourism Images.<br>Integrated Theme 5: The Role of Multicultural Development in Shaping Tourism Image.<br>Integrated Theme 6: Satisfaction with Tourism Infrastructure. | Theme 1: Language Proficiency and Multilingual Services to Attract Chinese Tourists.<br>Theme 2: Promoting Local Culture to Attract Chinese Tourists.<br>Theme 3: Lack of Competitive Advantage in Natural Resources.<br>Theme 4: Marketing and Product Innovation for the Chinese Market.<br>Theme 5: Service Excellence in Accommodation and Catering.<br>Theme 6: Maintaining Satisfaction in Healthcare and Hygiene.<br>Theme 7: In-depth Religious Experience for Chinese Tourists.<br>Theme 8: Sustaining Satisfaction with Infrastructure and Service Quality.<br>Theme 9: Tackling Regional Disparities.<br>Theme 10: Chinese Tourists' Preferences: Traditional Experiences Over Emerging Tourism. |
| | B | Temporary Theme B1: Enhancing Chinese language communication skills boosts competitiveness in the tourism sector, addressing the specific needs of Chinese tourists.<br>Temporary Theme B2: Language proficiency is linked to the improvement of tourist experiences and images, emphasizing the importance of effective communication in service quality.<br>Temporary Theme B3: Chinese tourists are generally satisfied with the natural and cultural tourism offerings, indicating successful alignment with their preferences.<br>Temporary Theme B4: Chinese tourists often lack a complete understanding of local and religious cultural images, pointing to potential areas for educational enhancement in tourism experiences.<br>Temporary Theme B5: Satisfaction with tourism infrastructure reflects well on destination management, highlighting the importance of maintaining high standards in facilities and services. | | |

**Table A1.** *Cont.*

| Perspective | Group | Searching for Themes | Reviewing Themes | Defining and Naming Theme |
|---|---|---|---|---|
| Attrac-tiveness | A | Temporary Theme A7: Language support plays a pivotal role in enhancing the tourism experience, enabling deeper engagement and enjoyment of local cultures. Temporary Theme A8: Language barriers impede the thorough understanding and experience of local cultures, underscoring the need for effective communication aids and services. Temporary Theme A9: Regional disparities in development affect overall attractiveness, suggesting a need for balanced investment in tourism infrastructure across different areas. Temporary Theme A10: A sense of unfamiliarity with local cultures among tourists indicates opportunities for cultural immersion and education within tourism offerings. Temporary Theme A11: A shortage of tourism service personnel affects the quality of accommodation and dining experiences, highlighting a critical area for workforce development. Temporary Theme A12: The allure of Islamic culture presents unique tourism opportunities, suggesting potential for specialized cultural tours and experiences. | Integrated Theme 7: The Comprehensive Role of Language Services in Enhancing Travel Experiences and Cultural Understanding. Integrated Theme 8: The Dual Role of Language Barriers and Services in Target Market Engagement. Integrated Theme 9: Regional Development Disparity and Its Impact on Tourism Attractiveness. Integrated Theme 10: The Fascination with and Unfamiliarity of Local Cultures. Integrated Theme 11: The Impact of Service Personnel Shortage on Hospitality Experience. Integrated Theme 12: Emerging Tourism Products and Market Preferences. Integrated Theme 13: Shopping Environment and Everyday Life Interest among Chinese Tourists. | Theme 1: Language Proficiency and Multilingual Services to Attract Chinese Tourists. Theme 2: Promoting Local Culture to Attract Chinese Tourists. Theme 3: Lack of Competitive Advantage in Natural Resources. Theme 4: Marketing and Product Innovation for the Chinese Market. Theme 5: Service Excellence in Accommodation and Catering. Theme 6: Maintaining Satisfaction in Healthcare and Hygiene. Theme 7: In-depth Religious Experience for Chinese Tourists. Theme 8: Sustaining Satisfaction with Infrastructure and Service Quality. Theme 9: Tackling Regional Disparities. Theme 10: Chinese Tourists' Preferences: Traditional Experiences Over Emerging Tourism. |
| | B | Temporary Theme B6: Chinese language services play a central role in attracting Chinese tourists, underscoring the importance of catering to language preferences in international tourism. Temporary Theme B7: Chinese-speaking guides are key in facilitating cultural understanding and enhancing the tourist experience, bridging the gap between different cultures. Temporary Theme B8: Chinese tourists show limited interest in emerging tourism products such as medical tourism, indicating a need for market research and product diversification. Temporary Theme B9: The dissatisfaction among Chinese tourists with the shopping environment in Malaysia points to areas for improvement in retail services and facilities. Temporary Theme B10: Multiculturalism is a key factor in attracting Chinese tourists, emphasizing the value of cultural diversity in tourism marketing strategies. Temporary Theme B11: Chinese tourists express interest in the everyday life of locals in non-touristic cities, indicating a desire for authentic and immersive experiences. | | |

**Table A1.** *Cont.*

| Perspective | Group | Searching for Themes | Reviewing Themes | Defining and Naming Theme |
|---|---|---|---|---|
| Competitiveness | A | Temporary Theme A13: Multilingual services are key to enhancing international tourism competitiveness, ensuring accessibility and inclusivity for a global audience.<br>Temporary Theme A14: Compared to neighboring countries, Malaysia's culture is more readily accepted by Chinese tourists, highlighting the importance of cultural compatibility in tourism attraction.<br>Temporary Theme A15: High levels of destination management contribute to a positive tourism experience, reflecting well on organizational and operational efficiency.<br>Temporary Theme A16: Compared to neighboring countries, cost is not Malaysia's advantage in tourism, suggesting a focus on value-added experiences to remain competitive.<br>Temporary Theme A17: In comparison to neighboring countries, natural tourism resources are not Malaysia's strength, indicating a need for innovation in other tourism sectors. | Integrated Theme 14: The Central Role of Multilingual Services and Professional Language Skills in Enhancing the International Competitiveness of the Tourism Industry.<br>Integrated Theme 15: Cultural Compatibility and Destination Management as Competitive Advantages.<br>Integrated Theme 16: Comparative Disadvantages in Cost and Natural Resources.<br>Integrated Theme 17: Marketing and Product Development Challenges.<br>Integrated Theme 18: Competitiveness in Accommodation and Catering Services.<br>Integrated Theme 19: Satisfaction with Local Healthcare and Hygiene Conditions. | Theme 1: Language Proficiency and Multilingual Services to Attract Chinese Tourists.<br>Theme 2: Promoting Local Culture to Attract Chinese Tourists.<br>Theme 3: Lack of Competitive Advantage in Natural Resources.<br>Theme 4: Marketing and Product Innovation for the Chinese Market.<br>Theme 5: Service Excellence in Accommodation and Catering.<br>Theme 6: Maintaining Satisfaction in Healthcare and Hygiene.<br>Theme 7: In-depth Religious Experience for Chinese Tourists.<br>Theme 8: Sustaining Satisfaction with Infrastructure and Service Quality.<br>Theme 9: Tackling Regional Disparities.<br>Theme 10: Chinese Tourists' Preferences: Traditional Experiences Over Emerging Tourism. |
| | B | Temporary Theme B12: The importance of Chinese language services in enhancing tourism competitiveness is evident, catering to the growing market of Chinese-speaking visitors.<br>Temporary Theme B13: Fluent Chinese-speaking guides play a role in boosting individual and industry competitiveness, enhancing personal and professional service quality.<br>Temporary Theme B14: The promotion of Malaysian tourism products to Chinese tourists lacks necessary business acumen, pointing to a need for strategic marketing approaches.<br>Temporary Theme B15: The lack of surprise among Chinese tourists at Malaysia's natural beauty suggests the need for innovative presentation and marketing of natural attractions.<br>Temporary Theme B16: Accommodation and dining services are competitive compared to neighboring countries, indicating strengths in hospitality offerings.<br>Temporary Theme B17: Satisfaction with local medical and health conditions reflects positively on the destination's public health infrastructure, contributing to overall tourist well-being. | | |

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
