# Peer review of "Unveiling the Drivers of Chinese Tourists’ Visit Intentions Regarding Malaysia"

_sustainability, doi:10.3390/su16083406_

Round 1

Reviewer 1 Report

Comments and Suggestions for Authors

It is good to see that the Chinese are good and desirable tourists:

"China has notably ascended as a global leader in the outbound tourism market, leading in both travel frequency and spending (Dichter et al., 2018)."

I think the above deserves greater emphasis in the text - to justify the importance of the study. 

I would have liked to have seen PRISMA being used for the systematic literature review. Please visit http://www.prisma-statement.org for more information. Thank you. 

I would have liked to have seen some reference to data triangulation in the methodology section. 

Additional comments:

1) The article is well written. Congratulations. 

2) The authors do need to add a clear research question, however, please.

Such as:

What are the reasons that have led Chinese tourists to choose to visit Malaysia less intensely, as a touristic destination? 

3) The relevance of the article is not clear. It needs to be clarified.

I perceive a table to be lacking justifying the relevance of the research gap, for example. Research question(s) in one column and references justifying them in (an)other column(s).

4) The authors do state:

"China has notably ascended as a global leader in the outbound tourism market leading in both travel frequency and spending (Dichter et al., 2018)."  

Albeit this is not enough. Many readers will not be enticed to read on or motivated to see what the article is about - as it currently stands.

Additionally, is not the preferred tourist to Malaysia from Singapore? And not from China?

Singapore is closer and they spend a lot of money on holiday in Malaysia as their currency is stronger (Malaysia is cheap for visitors from Singapore)...

What is the relevance of the Chinese tourist?

I see no such considerations, although Singapore is briefly mentioned in the text... (on p.2). 

5) The sample size and quality seems appropriate. But for what purpose? 

The methodology followed seems acceptable. But not very ambitious.

A reader may be led to think, concerning this research - "So what?"

This indeed was my first reaction. 

6) The above especially needs to be emphasised in the case of Western readers and researchers. Who may not be motivated to read the full article... They are more distant from destinations such as Malaysia and perhaps care less about what is going on there and who the incoming tourists are.

7) Is the article about fake news? I see at least one relevant reference in the references section to that end but this is not clear - nor is there a literature review about fake news in the text. This literature review hence needs adding to. 

8) The study lacks replicability - as the PRISMA protocol was not followed for the literature review. This needs to be corrected, to clarify some research design decisions - what is in the literature review and what is not? Why? 

9) The title is too long (e.g., more than 12 words...) and does not raise the readers' interest. Is doing semi-structured interviews a distinguishing factor of the article? I think not. Is news to be confused with scientific research? I do not think that to be the message either. The title needs rewriting.

10) The authors state in the conclusion that:

"Fully understanding the factors that influence tourists' intention to visit is key to increasing visitor numbers"

This is not new. The STP model - or segmentation, targeting, and positioning - has been followed for decades now. What is the real contribution of the article, to theory and to practice?

11) Additionally - the authors state in the conclusion:

"We find that the lack of intention to visit among Chinese tourists is not primarily caused by the negative news widely reported in Chinese media." This never occurred to me, anyway, so I do not see the relevance. Is there fake news being spread regarding this topic? How? In which channels? And why? What is the motivation for that?Fake news is a huge topic and the article suggests to the reader that fake news is occurring... But no in-depth analysis is done on this matter.

Comments on the Quality of English Language

Minor English imprecisions on occasion. 

Reviewer 2 Report

Comments and Suggestions for Authors

The paper "Beyond news impact: Employing semi-structured interviews to analyse factors affecting Chinese tourists' intentions to visit Malaysia" proposed a topic of study of interest to academia and the tourism sector, particularly Malaysia.

In general, it is considered that the topic addressed in the study is potentially interesting and presents novelties for both academia and the industry. Thus, the research question is clearly presented along with the research gap that the study addresses. Also, the methodology adopted allows the research question to be answered, although it would be relevant to combine qualitative analysis (interviews) with quantitative methods to help avoid potential bias in the results. However, the results are presented objectively and precisely and point to important theoretical and practical contributions in relation to the research objectives formulated.

Thus, this study presents a topic of interest and relevance to academia and industry, pointing out techniques that can help attract more Chinese tourists to Malaysia, and the authors are to be congratulated for this work and for the relevance of the topic investigated. To summarise, the article is considered to present a relevant and current topic and is in a position to be accepted for publication.

Reviewer 3 Report

Comments and Suggestions for Authors

Thank you for the opportunity to read the article “Beyond news impact: Employing semi-structured interviews to analyze factors affecting Chinese tourists’ intentions to visit Malaysia”

The authors investigate the factors influencing the intentions of Chinese tourists to visit Malaysia. The authors highlight the low level of tourist flow from China to Malaysia as a problem that has not yet returned to the levels that existed before COVID-19. The authors aim to identify the factors influencing the intention of tourists to visit Malaysia. In addition, the article highlights the lack of adequate methods for studying these factors. However, the purpose of the article and the research questions posed by the authors are not clearly formulated. Although the title suggests that the article will focus on a specific method of describing the factors influencing the intentions of Chinese tourists to visit Malaysia, it is unclear whether this will be the main focus of the article. Then we learn that " research contributes to the academic field in several distinct ways." The authors should clearly formulate the purpose of the article, research hypotheses or research questions. The mention of the method should be removed from the title, since the main focus of the article is on the factors, not the method.

The literature review includes a wide range of research on various aspects of the topic, highlighting the gaps that define the specifics of this article. This confirms the relevance of the topic and the importance of this study. The authors provide detailed descriptions of the theoretical and practical implications in separate sections.

The methods and data processing are also described in detail with links to the articles on which the methodology of the authors is based.

The authors compare their findings with those of other researchers and emphasize the novelty of their findings.

The volume of the article is 33 pages, including 10 appendices and a table on three pages. It may be useful for authors to consider storing some of this data in the cloud, making it available on request.
